# Absence of Yield Reduction after Controlled Water Stress during Prehaverst Period in Table OliveTrees

**María José Martín-Palomo** [1,2], **Mireia Corell** [1,2], **Ignacio Girón** [2,3], **Luis Andreu** [1,2],
**Alejandro Galindo** [1,2], **Ana Centeno** [4], **David Pérez-López** [4] and **Alfonso Moriana** [1,2,*]

1   Departamento de Ciencias Agroforestales, Universidad de Sevilla, ETSIA, Crta de Utrera Km 1,
    41013 Seville, Spain; mjpalomo@us.es (M.J.M.-P.); mcorell@us.es (M.C.); landreu@us.es (L.A.);
    agegea@us.es (A.G.)
2   Unidad Asociada al CSIC de Uso Sostenible del Suelo y el Agua en la Agricultura (US-IRNAS),
    Crta de Utrera Km 1, 41013 Seville, Spain; iggi@irnase.csic.es
3   Consejo Superior de Investigaciones Científicas, IRNAS, Avda Reina Mercedes, 10, 41012 Seville, Spain
4   Departamento Producción Agraria, CEIGRAM-Universidad Politécnica de Madrid, Av. Puerta de Hierro, 2,
    28040 Madrid, Spain; ana.centeno@upm.es (A.C.); david.perezl@upm.es (D.-P.L.)
*   Correspondence: amoriana@us.es; Tel.: +34-54486456

**Abstract:** Deficit irrigation scheduling is becoming increasingly important under commercial conditions. Water status measurement is a useful tool in these conditions. However, the information about water stress levels for olive trees is scarce. The aim of this experiment was to evaluate the effect on yield of a moderate controlled water stress level at the end of the irrigation season. The experiment was conducted in the experimental farm of La Hampa (Coria del Río, Seville, Spain) during three years. A completely randomized block design was performed using three different irrigation treatments. Deficit irrigation was applied several (4 or 2) weeks before harvest. Irrigation was controlled using the midday stem water potential, with a threshold value of −2 MPa and compared with a full irrigated treatment. This water stress did not reduced gas exchange during the deficit period. The effect on yield was not significant in any of the three seasons. In the high-fruit load season, fruit volume was slightly affected (around 10%), but this was not significant at harvest. Results suggest an early affection of fruit growth with water stress, but with a slow rate of decrease. Moderate water stress could be useful for the management of deficit irrigation in table olive trees.

**Keywords:** fruit size; Manzanilla; olive; regulated deficit irrigation; water potential; water relation

## 1. Introduction

Olive orchards grow around the world in semi-arid or arid conditions, with great water scarcity. Deficit irrigations are very common in these areas. Sometimes, growers are not aware that their water management may cause water stress periods. The yield response to water stress conditions is related to the duration, the level and the moment when the plant water status is affected [1]. Although olive trees are considered extremely resistant to drought conditions [2], the full bloom/fruit set period [3] and the oil accumulation [4] have been considered sensitive to irrigation shortages. On the other hand, the pit hardening has been traditionally considered the most drought-resistant phenological stage [5], and irrigation restrictions are commonly scheduled in this period. Irrigation scheduling for table olive cultivars is more complex than for oil ones, especially in the case of green olives. The main limitations of regulated deficit irrigation (RDI) for table cultivars are the importance of the fruit size in the final value of the yield and the short period available for rehydration. In addition, the harvest period for green table olives occurs at the end of the irrigation season, when there is a very small amount of irrigation water available and rains are scarce.

RDI scheduling was defined using the drought resistant and sensitive periods, when irrigation shortages could or could not be performed [6]. Some of these sensitive periods occur at the end of the season, during fruit ripening or in the last fruit growth stages. A moderate water stress before harvest has been linked to a flavour improvement of the fruit (in tomato, Ref. [7]; in peaches, Ref. [8]) although it usually causes a reduction of fruit yield (peaches, Ref. [9]). Reference [10] discussed several irrigation works for peach trees where the RDI strategies results are affected by the water holding capacity of the soils. In the review, similar irrigation restrictions produced different results [10]. On the other hand and also for peach trees, Ref. [8] reported the absence of significant differences in yield between full irrigation and water stress during stage III of fruit development, when the water stress level was controlled. Both works suggested that if the water stress level is controlled during drought-sensitive phenological periods, such as stage III for peaches, a reduction of irrigation could be applied without any yield reduction. This benefit arising from moderate water stress conditions is also reported in olive trees. Water stress conditions increased total amount of phenols in fruit with significant changes in the olive oil flavour [11]. These results would explain the reduction of susceptibility to bruising in cv Manzanilla suggested by the water stress in this period [12]. Moderate water stress conditions have also been linked to an improvement of the oil accumulation rate [3,13]. Therefore, the final growth and ripening phases in olive trees could be not as sensitive as initial works suggested. For table olive trees, Ref. [14] reported that midday stem water potential values higher than −2 MPa did not affect fruit growth in cv Manzanilla. More recently, Ref. [15] also suggested this value as a possible threshold for olive irrigation scheduling. The tree response to this level of water stress (−2 MPa) could be different if the irrigation restriction was performed later in the season, or if no rehydration was considered. The aim of this work is to study the yield response of table olive trees to water restrictions before harvest, using −2 MPa of midday stem water potential as threshold value. In addition to water stress level, the duration of the stress was also considered in the evaluation of treatments.

## 2. Material and Methods

### 2.1. Site Description and Experimental Design

Experiments were conducted at La Hampa, the experimental farm of the Instituto de Recursos Naturales y Agrobiología (IRNAS-CSIC), located in Coria del Río near Seville (Spain) (37 17′N, 6 3′W, 30 m altitude). In them, 44-year-old table olive trees (*Olea europaea* L cv Manzanillo) were observed from the 2014 to the 2016 seasons. The tree spacing followed a 7 m × 5 m square pattern. The sandy loam soil (about 2 m deep) of the experimental site was characterised by a volumetric water content of 0.33 $m^3$ $m^{-3}$ at saturation, 0.21 $m^3$ $m^{-3}$ at field capacity and 0.1 $m^3$ $m^{-3}$ at permanent wilting point, and 1.30 (0–10 cm) and 1.50 (10–120 cm) g $cm^{-3}$ bulk density. Pest control, pruning and fertilization practices were those commonly used by growers, and weeds were removed chemically within the orchard, pruning was avoided only in the last season. Drip irrigation was carried out at night, using one lateral pipe per row of trees and five emitters per plant, spaced 1m and delivering 8 L $h^{-1}$ each. There were problems with the irrigation system at the beginning of the experiment (2014 season) that reduced water applied in some plots. However, such reduction did not affect plant water relations.

The experimental design was a completely randomized block experiment with 3 blocks and 3 irrigation treatments. Each treatment was carried out in a plot with two trees aligned in a single row and two adjacent guard rows. The amount of water was measured using a water meter for each plot. Control trees were irrigated in order to obtain the optimum tree water status throughout the season. The midday stem water potential (see below for more details) was used to estimate the water stress level. All treatments were irrigated in the same way as the Control treatment from the end of the spring to the beginning of summer, depending on the water status of the trees. Each repetition was scheduled independently. The optimum values for Control treatments were −1.2 MPa before and −1.4 MPa after the beginning of massive pit hardening [16]. The beginning of the massive pit hardening period was estimated according to [17] and the date was determined when a change in the

slope of the longitudinal fruit growth was measured. This period started around mid-June, 17, 10 and 15 June, respectively in 2014, 2015 and 2016. The irrigation restrictions were applied according to the estimated harvest date (around mid-September), starting approximately four weeks before harvest (RDI 2) and two weeks before harvest (RDI 1). The beginning and the end (harvest) of the treatment were changed according to the fruit load and fruit development in each season. Harvest date occurred on 15 (2014 season), 3 (2015 season) and 27 (2016 season) September. Deficit trees were irrigated during this period only if the midday stem water potential measurements were below −2.0 MPa [14]. No irrigation was performed after harvest for any of the treatments.

The irrigation needs of individual plots were changed weekly depending on the distance to the threshold value considered [14]. Three levels of irrigation rate were estimated based on the maximum average daily crop evapotranspiration (ETc) of the orchard (4 mm day$^{-1}$). This estimation was calculated for the last ten years with the Kc and Kr recommended values [18]. The irrigation rate varied according to Table 1.

**Table 1.** Applied water was estimated according to the comparison of the measured midday stem water potential and the threshold considered. Irrigation was provided only when measured values were lower than threshold.

| % of Decrease from the Threshold | Amount of Irrigation (mm day$^{-1}$) | % of Average Maximum ETc |
|---|---|---|
| Less than 15% | 1 mm | 25% |
| Between 15% and 30% | 2 mm | 50% |
| Greater than 30% | 4 mm | 100% |

*2.2. Meteorological Conditions throughout the Experiment*

Weather data during the three seasons were obtained from "La Puebla" station in the Andalusian weather stations network (Andalusian weather stations network. SIAR, Spanish Agriculture, Fish and Food Spanish Ministry). This station is around 5 km away from the experimental orchard, and data are available at SIAR web page (http://eportal.mapama.gob.es/websiar/SeleccionParametrosMap.aspx?dst=1). During the three seasons, the patterns of daily potential evapotranspiration (ETo) and rainfall events were similar to the average season (Figure 1). The ETo reached a peak at the end of spring-beginning of summer period with maximum values near 8mm day$^{-1}$; the average of summer data was around 6mm day$^{-1}$. Minimum ETo values were measured during winter, with values around 1mm. Rain was almost null from the end of spring until end of summer throughout the experiment. The rainy period concentrated in autumn and winter, as usual in Mediterranean climates, with 80% of the total amount of rain in 2014 and 2015 and around 60% in 2016. There were some rainy episodes a few days before harvest (2 mm from 7 to 12 September in 2014 and harvest was on 15 of September, 11.5 mm on 13 September in 2016 and harvest was on 22 of September). According to the average season (539 mm, Ref. [19]), the year 2015 was very dry, with only 289 mm of rain, and this increased the amount and duration of the irrigation. Conversely, the 2016 season was slightly wetter than average, with 643 mm and an important increase of rain events in Spring.

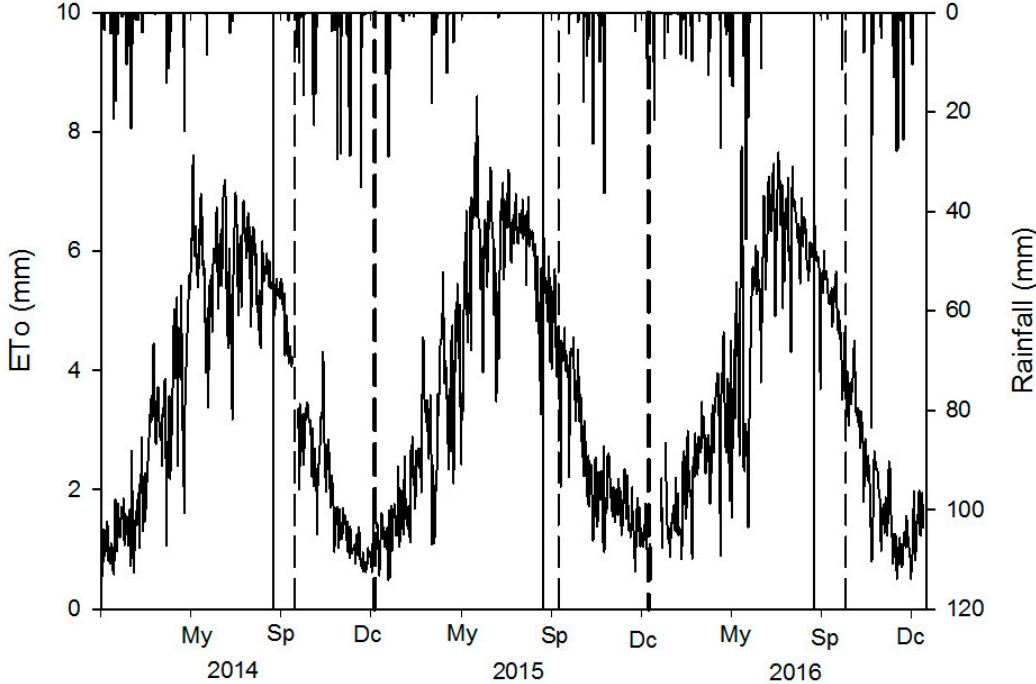

**Figure 1.** Pattern of daily potential evapotranspiration (ETo, mm) and rainfall (mm) during the three seasons of the experiment (2014 to 2016). Vertical bold dash lines separate each season. Dates between vertical solid and dash lines in each year represent the deficit period. Based on data from "La Puebla" weather station (Andalusian weather stations network). Data download from SIAR, Agriculture, Fish and Food Spanish Ministry (http://eportal.mapama.gob.es/websiar/SeleccionParametrosMap.aspx?dst=1).

*2.3. Measurements*

Irrigation strategies were characterized using several water relations measurements. A locally calibrated portable FDR (HH2, Delta T, UK) was used to obtain soil moisture measurements. The measurements were made in three plots per treatment. The access tubes for the FDR sensor were placed in the irrigation line, at about 30 cm from the nearest emitter [20]. Data were obtained at 1 m depth and 10 cm intervals. These data were used to estimate the relative extractable water (REW) using the Equation (1).

$$REW = (R - Rmin)/(Rmax - Rmin),\tag{1}$$

Where:
R: Actual soil water content
Rmin: Minimum soil water content measured in the experiment (0.1 $m^3$ $m^{-3}$)
Rmax: Soil water content at field capacity. Estimated as approximately 0.21 $m^3$ $m^{-3}$

The leaf gas exchange was measured using two different methods. Daily cycle of leaf conductance in olive trees presents a maximum value during the morning and decreases until minimum plateau at midday [21]. Maximum values have been reported as earlier indicators for detecting water stress [22]. Then, during the 2014 and 2015 seasons, the abaxial leaf conductance was measured at around 10:00 a.m. in order to estimate the maximum daily value in two fully expanded sunny leaves per tree with a steady state porometer (SC-1, Decagon devices Inc., Pullman, WA, USA). However, differences between treatment were lower than expected. This lack of results could be related with the time of the measurement because the period of maximum values could be shorter tan expected. Then, in the last season, a more sensitive method was used during midday when leaf conductance values were at steady state. During 2016, the midday leaf net photosynthesis was measured with an infrared gas analyser (CI-340, CID BioScience, Camas, USA) in two fully expanded sunny leaves per tree. The water potential was measured weekly at midday in one leaf per tree, using the pressure chamber

technique [23]. The leaves near the main trunk were covered in aluminium foil at least one hour before measurements were taken, and a pressure bomb was used for the measurements (PMS model 1000). In order to describe the cumulative effect of the water deficit, the water stress integral (SI) was calculated based on the midday stem water potential data (Equation (2), Ref. [24]) from the beginning of pit hardening until harvest and from the beginning of the irrigation restriction. The maximum value suggested by [24] was changed with the same reference value of −1.4 MPa, which is the threshold value suggested by [16] in fully-irrigated olive trees. Any water potential value higher than the reference was considered as equal to this. The expression used was:

$$\text{SI} = |\sum (\text{SWP} - (-1.4) * n)|, \tag{2}$$

where SI is the stress integral; SWP is the average midday stem water potential for any interval; $n$ is the number of days in the interval.

At the beginning of each season, ten shoots per tree were selected randomly. The number of fruits were measured periodically in the morning for each shoot. The fruit volume was estimated from a survey of ten fruits per tree. Fruits were randomly selected on each date of measurement. Two measurements were made for each fruit: the longitudinal dimension and the transversal dimension (at the equatorial point) and used both for estimated the volume of an ellipsoid. The pattern of the longitudinal fruit growth was used to estimate the beginning of the pit hardening [17].

The irrigation treatments were also evaluated from the point of view of quantity and quality of yield. For table olives, the quality of the fruit is related to several parameters; some of the most common in the industry are the pulp/stone (PS) ratio, fruit size, fruit colour and fruit firmness. High values of pulp/stone (PS) ratio are considered an indicator of better-quality fruits. The pulp-stone ratio was measured in fresh and dry weight. Pulp and stone of 30 fruits per treatment (10 fruits per plot) were separated and weighed while fresh. Then, these samples were dried at 60 °C until the weight became constant; then, they were weighed again. The final fruit size was estimated in 6 trees per treatment using the number of fruits per kilogram. A sample of around 500 fruits per tree was counted and weighed. Fruit colour is also an important feature for green table olive trees. Spots in the fruit due to the beginning of ripening reduce the yield value. The fruit colour was evaluated using the mature index [25], applied to a sample of 50 fruits that, according to the spots, were ranked with 0 for green fruit, 1 for yellow-green fruit (optimum for green table olives), 2 for purple spots in less than the 50% of the fruit, 3 for purple spots in more than 50% of the fruit and 4 for 100% purple fruit. Each sample was evaluated using the average weight, number of fruits and marks. Fruit firmness was defined as the maximum force required to compress a sample; more specifically, the peak force of the first compression of the fruits [26] was measured using a penetrometer (PCE FM 200, Albacete, Iberica) in 10 fruits per plot. Crop water productivity (CWP) was estimated as the ratio between fruit yield and water consumed in each plot [27]. The water consumed was estimated as the sum of the total water applied and the rainfall during the irrigation period, because the rest of components in the water balance are considered negligible [28]. Only drainage could be considered significant during some periods. Drainage was detected using the comparison of consecutive soil moisture data measurements in the deeper horizon (1m). The data indicating drainage was lower than 10%, and some of them were related to previous rainfall events, only during 2015, when this percentage increased slightly to 15%. Since some works reported olive root growth below 1 m depth (i.e., Ref. [3]), drainage was considered negligible.

Data analyses were performed using ANOVA, and the mean separation was made using Tukey's test with the Statistix (SX) program (8.0). Significant differences were considered when $p$-level < 0.05 in both tests. Calculations of the $p$-level were performed considering the $F$-test of variance equality. When conditions of variance equality were not obtained, a decrease in the degree of freedom and, therefore, more restrictive $p$-values were calculated. The number of samples measured is specified in the text and figures. The relationship between relative fruit volume and water potential was calculated

using all individual data available. The relative fruit volume was calculated considering the maximum value as 100% for each season. The enveloping curve for all data was estimated as the regression of percentile 75% and intervals of 0.5 MPa (SWP) and 5 MPa × day (SI).

## 3. Results

The water pattern applied during the three years of the experiment is shown in Figure 2. The irrigation season lasted from mid-spring until the end of summer, around mid-September, when trees were harvested. This was the dry period of the season, although rainfall events were measured at the end of the 2014 and 2016 seasons (Figure 1).

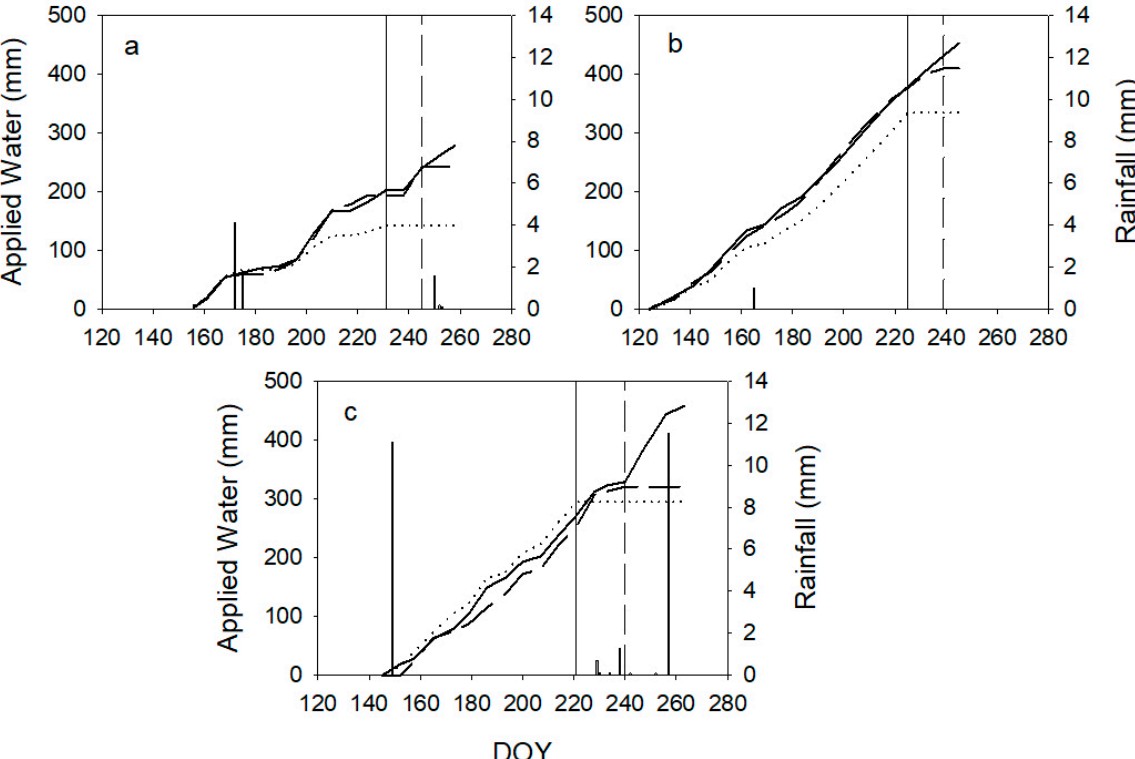

**Figure 2.** Water applied (lines) and rainfall (bars) during the irrigation season in the three years of the experiment (**a**) 2014, (**b**) 2015, (**c**) 2016. Solid lines, control treatment; dash line, regulated deficit irrigation (RDI) 1 treatment; and dotted line, RDI 2 treatments. Values are the average of three data. The vertical solid line indicates the beginning of the RDI 2 treatment. The vertical dash line indicates the beginning of the RDI 1 treatment.

The pattern of water applied is similar for the different seasons, with almost a linear increase in all treatments until the period of irrigation reduction. The irrigation was scheduled based on water potential measurements and these produced slight changes in some treatments before the irrigation restrictions. The maximum amount of water used corresponded to the 2015 season (Figure 2b), a low fruit load year, because of the scarce rainfall in spring and summer responsible for advancing the beginning of irrigation (around one month before than the rest of the seasons). The moment to start the irrigation restriction was different in each year because the harvest date varied. The shortest water stress period took place in 2015 (20 days in RDI 2 and 6 days in RDI 1, Figure 2b) because the low fruit load brought forward the beginning of the ripening and, therefore, the harvest. Conversely, the longest period happened in 2016 (42 days in RDI 2 and 23 days in RDI 1, Figure 2c) because this season was the one with the highest yield.

The soil moisture pattern is showed in Figure 3 using relative extractable water (REW). During the 2014 season, there were significantly lower values in RDI 2 compared to the rest, before treatments

started. Such results were likely related to problems with irrigation during the season. These differences were not found in the following two seasons, when the values of the three treatments were similar before irrigation restrictions.

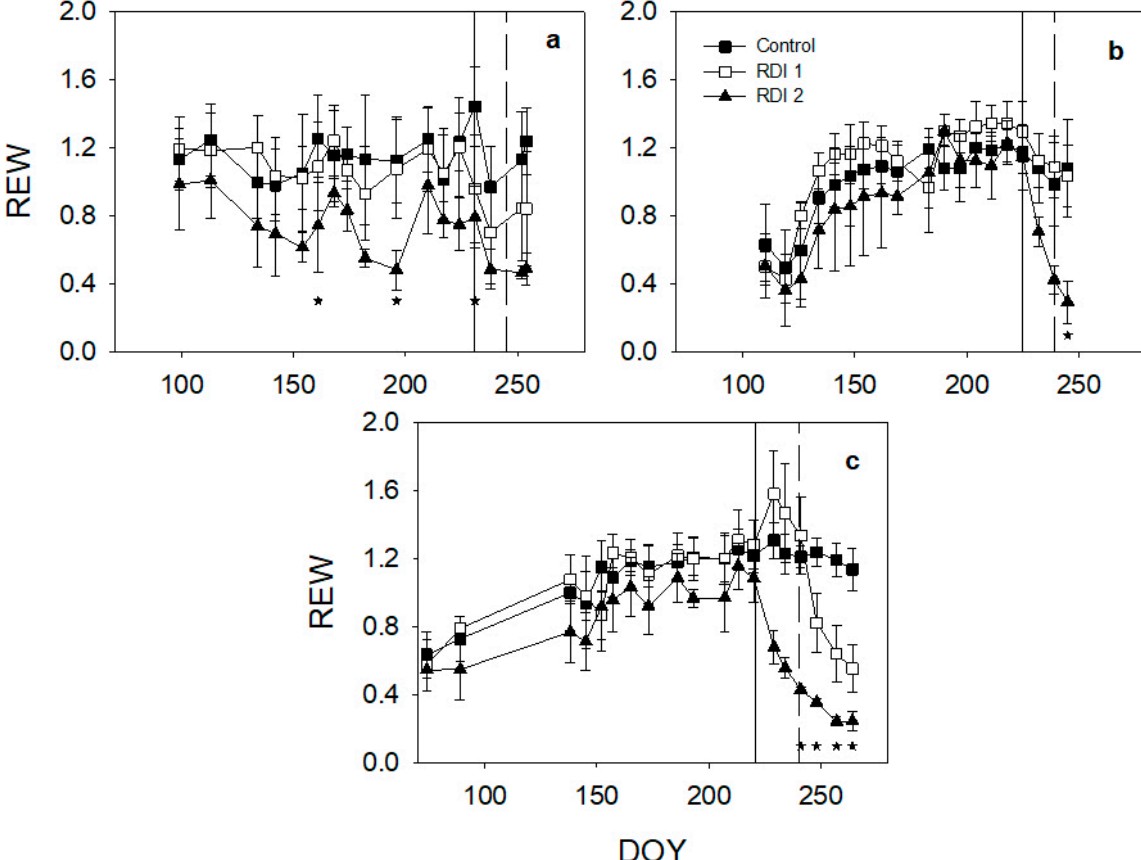

**Figure 3.** Relative extractable water (REW) at 1 m depth throughout the experiment during the 2014 (**a**), 2015 (**b**) and 2016 (**c**) seasons. Solid squares are Control treatments; empty squares are RDI 1; and triangles are RDI 2 treatments. Values are the average of three data. The vertical solid line indicates the beginning of the RDI 2 treatment. The vertical dash line indicates the beginning of the RDI 1 treatment. Asterisks show when significant differences were found. (Tukey Test, $p < 0.05$).

There were no significant differences in the dry period of the 2014 and 2015 seasons, only in the last data of RDI 2 in 2015. However, soil moisture decreased clearly in RDI 2 trees after the beginning of the treatments. On the contrary, the 2016 season had the longest dry period, and significant differences were found between RDI 2 and Control from DOY 242, and between RDI 1 and Control from DOY 258. RDI 1 and RDI 2 were significantly different only in the period 242–249 in this latter season. Values below 0.4 were measured only in RDI 2, only at the end of the 2015 season and in the last three weeks of 2016. REW data were slightly higher than 1 on most of the dates; only at the beginning of the season in 2015 and 2016 values around 0.4 were found in all treatments.

Irrigation was scheduled based on the midday stem water potential (SWP) in the three seasons (Figure 4).

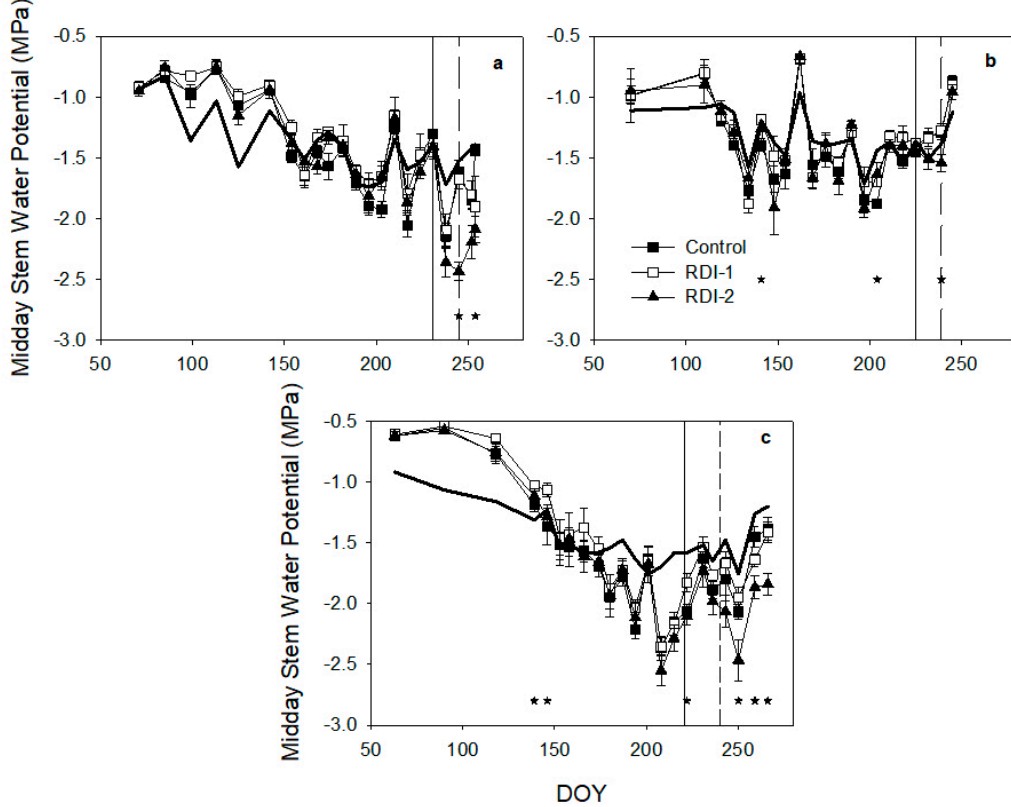

**Figure 4.** Midday stem water potential throughout the experiment during the 2014 (**a**), 2015 (**b**) and 2016 (**c**) seasons. The solid line represents the baseline calculated according to Corell et al. (2016). Values are the average of six data. The vertical solid line indicates the beginning of the RDI 2 treatment. The vertical dash line indicates the beginning of the RDI 1 treatment. Asterisks show when significant differences were found. (Tukey Test, $p < 0.05$).

Before the beginning of the irrigation restriction, most data were almost equal; the seasonal pattern was also similar between treatments and close to the baseline of [29]. Only during 2015 and 2016, there were two significant differences before the period of restriction. Such differences were small and isolated during the experiment. In 2016, a very high-fruit-load season, all treatments reduced the SWP values. The baseline also predicted this SWP pattern, so the evaporative demand was, in part, related to this response. However, the SWP values in the period 208–222 were even lower than the baseline, and moderate water stress conditions could occur. During the deficit period, RDI 2 SWP decreased around 2 weeks after the irrigation restriction in 2014 and 2016 (Figure 4a,c); only in the low fruit load season, it was almost null (2015, Figure 4b). Such delay was likely related to a not limited capacity of the trees to obtain soil water, probably because the reduction of evaporative demand (Figure 1) increased the percentage of water available in the soil. Only during 2016, there was a long clear period of around three weeks with significant lower SWP in RDI 2 than in Control. In RDI 1, these differences were even lower and not significant. In all the seasons, the recovery for all the treatments was related to rainfall events. Minimum values of SWP, around −2.5 MPa, were measured in RDI 2 during the high fruit load season (2014 and 2016).

The stress integral (SI) was not significantly different between treatments during the low-fruit-load season (2015, Figure 5), when the lowest SI values were calculated. During 2014, no significant differences were found when the overall season was considered. However, in the deficit period, significant differences in SI were found. The RDI 2 value was approximately double (20 MPa·day) that of Control and RDI 1 (approximately 10 MPa·day). In the 2016 season, the SI values from the beginning of pit hardening were the highest, around two-fold higher than the ones from 2014 and 50% higher

when the RDI 2 irrigation restriction was considered. RDI 2 was significantly higher than RDI 1 but not higher than Control, from pit hardening in 2016. However, it was significantly higher than the other two treatments when only the period of restriction is considered.

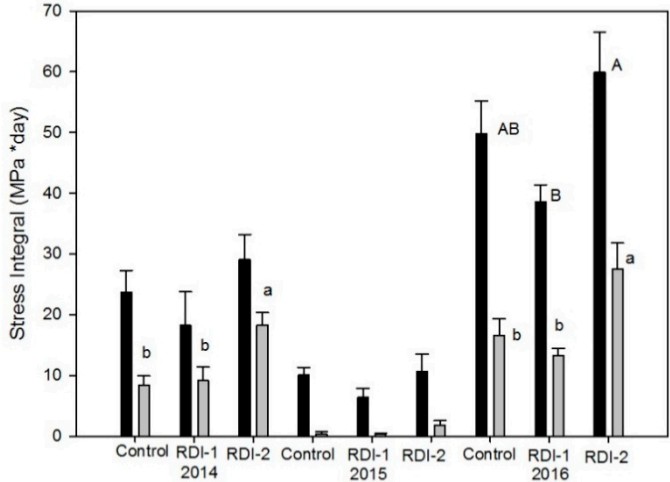

**Figure 5.** Stress integral (SI) during the three seasons of the experiment. Black bars represent the SI from the pit hardening phase until harvest (2014 DOY 168–254, 2015 DOY 162–245, 2016 DOY 166–266). Grey bars show the SI values only from the RDI 2 irrigation restriction (2014 DOY 231–254, 2015 DOY 225–245, 2016 DOY 221–266). In the three seasons, the reference value to calculate the SI was −1.4 MPa. Each bar is the average of three data. Capital cases indicate significant differences in the season considered from the period of pit hardening. Lower cases show significant differences in the season considered only from the RDI 2 irrigation restriction. (Tukey Test, $p < 0.05$).

Table 2 summarises irrigation, water potential and the stress integral for the three seasons according to three different phenological phases. The first two phases considered (vegetative growth and pit hardening) were the periods when irrigation target was an optimum water status in all treatments. During the vegetative growth period, rains partially covered the water needs and the water status of the trees was typically around the threshold (−1.2 MPa). During pit hardening, the evaporative demand increased and rainfall was almost null, which increased the amount of irrigation in order to maintain the midday water potential around −1.4 MPa. For both periods in all seasons, water potential and stress integral values in all treatments were very similar. In the deficit period, rains were also very scarce but the amount of irrigation was lower because the duration was shorter than in previous periods. The water status of RDI 2 was worse than Control and RDI 1 in all seasons.

**Table 2.** Summary of irrigation amount (Irr, mm), average midday water potential (SWP, MPa) and stress integral (SI, MPa × day) per season and phenological stage. Each period is characterized using the average daily potential evapotranspiration (ETo, mm) and total rainfall (R, mm).

|  |  | **2014** | | |
|---|---|---|---|---|
|  |  | **Control** | **RDI 1** | **RDI 2** |
| **Vegetative Growth (71–168)** | **Irr** | 54 | 56 | 56 |
| **ETo 4.7/R 78.6** | **SWP** | −1.12 | −1.04 | −1.11 |
|  | **SI** | 2.3 | 1.1 | 1.6 |
| **Pit Hardening (169–230)** | **Irr** | 149 | 139 | 87 |
| **ETo 5.7/R 5.9** | **SWP** | −1.62 | −1.49 | −1.54 |
|  | **SI** | 17.5 | 11.6 | 14.2 |
| **Deficit Period (231–258)** | **Irr** | 75 | 47 | 0 |
| **ETo 4.9/ R 1.9** | **SWP** | −1.76 | −1.87 | −2.27 |
|  | **SI** | 6.2 | 6.6 | 14.9 |

**Table 2.** *Cont.*

|  |  | 2015 |  |  |
|---|---|---|---|---|
|  |  | **Control** | **RDI 1** | **RDI 2** |
| **Vegetative Growth (70–161)** | **Irr** | 134 | 124 | 105 |
| **ETo 4.7/R 55.9** | **SWP** | −1.28 | −1.22 | −1.25 |
|  | **SI** | 5.7 | 3.6 | 5.1 |
| **Pit hardening (162–225)** | **Irr** | 244 | 252 | 228 |
| **ETo 6.0/R 1.6** | **SWP** | −1.46 | −1.40 | −1.44 |
|  | **SI** | 9.9 | 6.2 | 9.5 |
| **Deficit Period (225–245)** | **Irr** | 74 | 33 | 0 |
| **ETo 4.9 /R 0.0** | **SWP** | −1.16 | −1.17 | −1.34 |
|  | **SI** | 3.2 | 1.8 | 10.7 |
|  |  | 2016 |  |  |
|  |  | **Control** | **RDI 1** | **RDI 2** |
| **Vegetative Growth (63–166)** | **Irr** | 62 | 63 | 73 |
| **ETo 4.1/R 253.9** | **SWP** | −1.14 | −1.02 | −1.12 |
|  | **SI** | 3.3 | 0.9 | 2.0 |
| **Pit hardening (167–221)** | **Irr** | 210 | 190 | 222 |
| **ETo 6.3 /R 0.0** | **SWP** | −1.98 | −1.90 | −2.01 |
|  | **SI** | 36.5 | 29.0 | 37.0 |
| **Deficit period (222–263)** | **Irr** | 185 | 67 | 0 |
| **ETo 4.8/R 13.9** | **SWP** | −1.70 | −1.70 | −1.99 |
|  | **SI** | 16.5 | 13.3 | 27.5 |

Figure 6 shows the gas exchange data. Significant differences were found in the three seasons, but they were only on isolated dates in the 2014 and 2015. Such results were not in agreement with the soil moisture and water potential, and they were likely not produced for the irrigation management. In the 2014 and 2015 seasons, the maximum leaf conductance was measured (Figure 6a,b). The leaf conductance was increasing throughout the experiment in both seasons but with a similar pattern for the different treatments. No clear trends were found between treatments. The midday net photosynthesis (Pn) was measured throughout 2016 (Figure 6c). Pn values were more similar during the season than those for leaf conductance. The significant differences measured from DOY 194–231 were not likely related to the irrigation treatments because water potential and soil moisture showed the opposite trend and, on all the dates, the Control treatment showed the lowest significant Pn value. Only the decrease in midday stem water potential on DOY 194 (Figure 4c) was coincident with a decrease in Pn (Figure 6c) in the 2016 season.

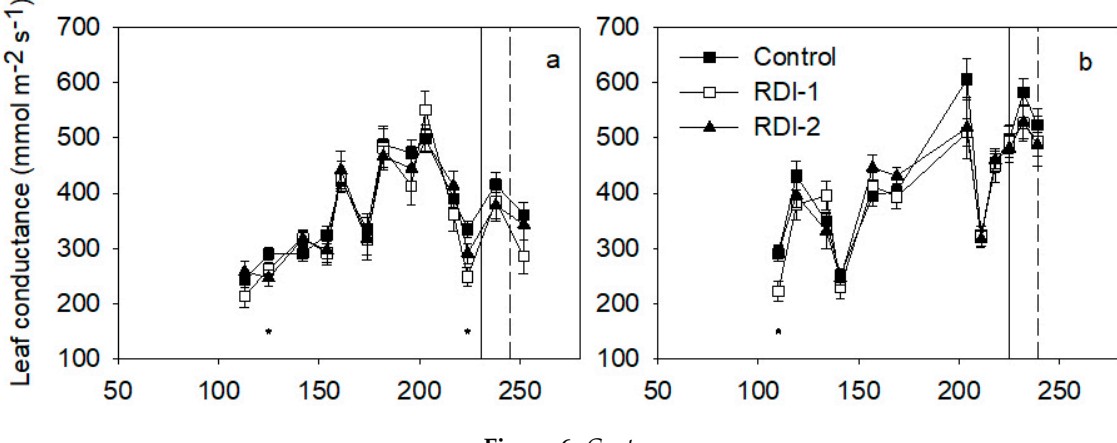

**Figure 6.** *Cont.*

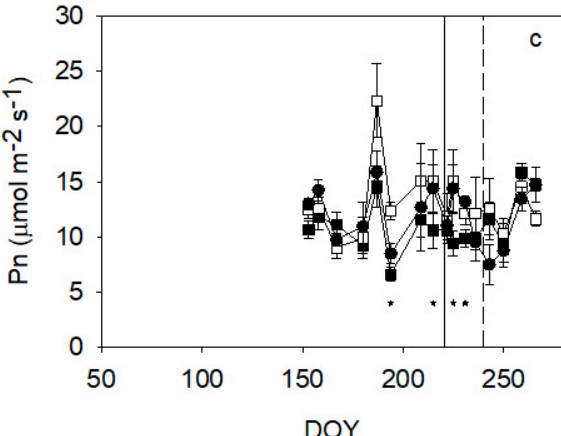

**Figure 6.** Gas exchange during the three seasons of the experiment. Maximum daily leaf conductance in the 2014 (**a**) and 2015 (**b**) season. Midday net photosynthesis (Pn) in 2016 (**c**). Solid squares are the Control treatments; empty squares are the RDI 1 treatments; and triangles are the RDI 2 treatments. Each point is the average of 6 data. The vertical solid line indicates the beginning of RDI 2 treatment. The vertical dash line indicates the beginning of RDI 1 treatment. Asterisks show when significant differences were found. (Tukey Test, $p < 0.05$).

The data of fruits per shoot presented clear differences between seasons (Figure 7). The fruit load in the 2015 season was almost null in all the treatments, while in 2016 the fruit load was extremely high. This pattern of alternate bearing was the same for the three treatments. Two sampling dates are presented for each season, at the beginning of massive pit hardening and at harvest. There were no significant differences between treatments within the year on the two sampling dates. The difference between both sampling dates (fruit drop) was also not significant.

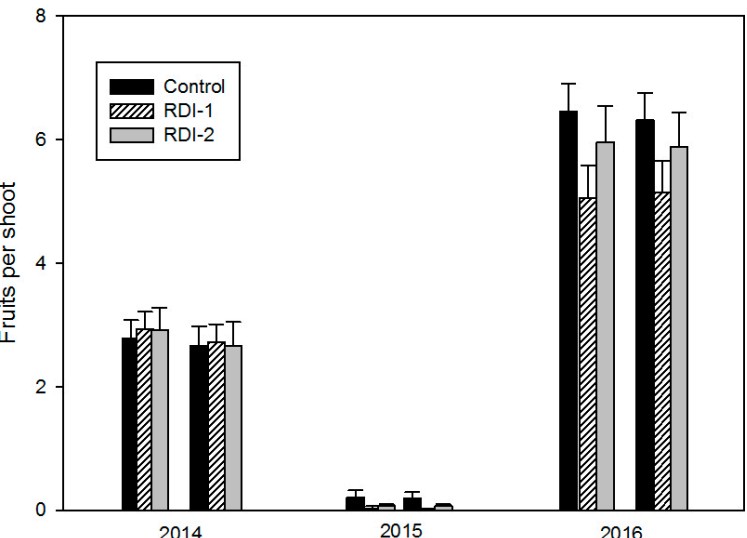

**Figure 7.** Amount of fruit peer shoot in the three seasons of the experiment. Black bars are the Control, white bars are the RDI 1 and grey bars are the RDI 2. Each bar is the average of 60 data. The vertical bar represents the standard error. For each season, bars on the left include the measurements at the beginning of the pit hardening and on the right the data before harvest. There were no significant differences between treatments in any of the seasons.

The pattern of fruit volume was almost linear before the beginning of the treatments in the three seasons (Figure 8). Although there were significant differences between treatments before these dates, they were small and unclear. The final fruit size was clearly related to the season; maximum values

were measured in 2015 (Figure 8b), the low fruit load year, and minimum values in 2016 (Figure 8c), the highest fruit load year. In the 2014 season, the deficit treatments presented a slight reduction in fruit volume during the stress period, but only in RDI 1 this was significantly lower than in Control. Rainfalls before harvest rehydrated all the treatments, and no differences were found in the end. In the 2015 season, the fruit growth was unaffected by the irrigation deficit and showed a linear increase of fruit volume. In 2016, the growth stopped after the deficit period in RDI 1 and RDI 2, while in Control it continued. Such differences were not very high, around 10%, and were reduced by the rainfall events just before harvest.

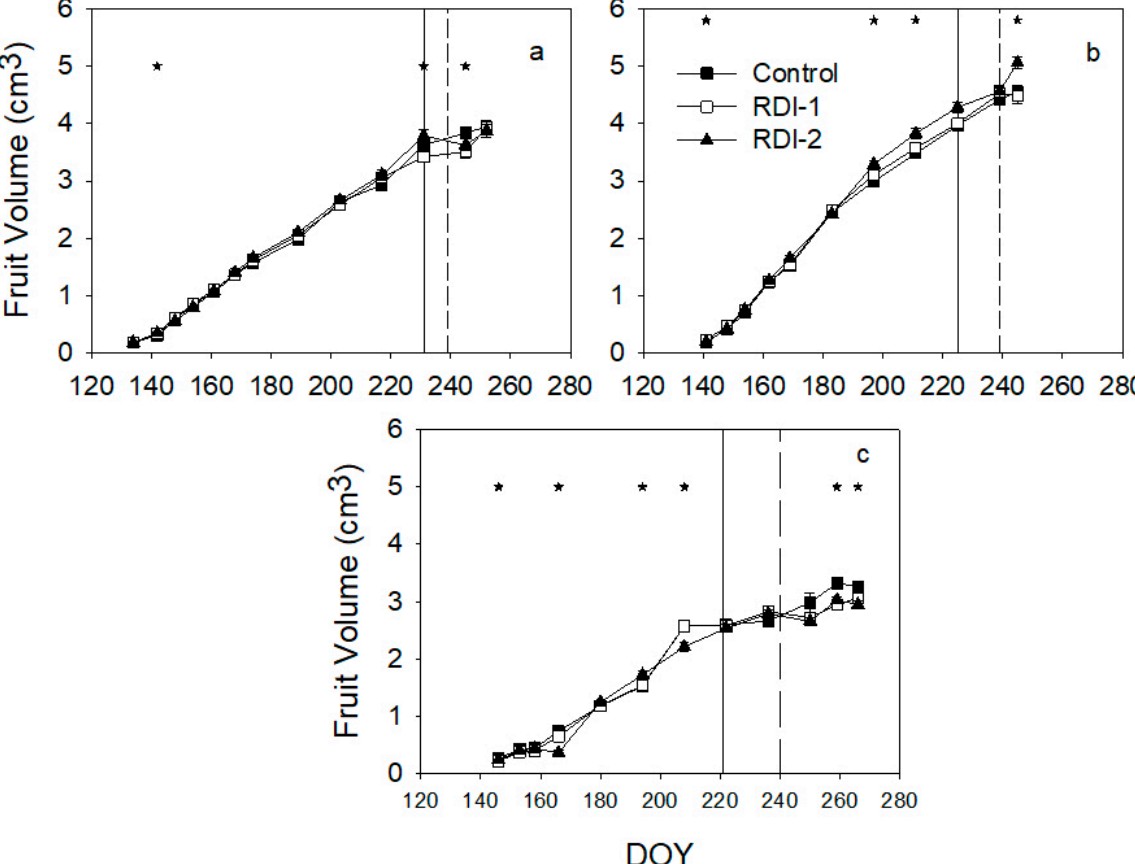

**Figure 8.** Patterns of fruit volume during the three seasons of the experiment (**a**) 2014, (**b**) 2015 and (**c**) 2016. Full squares are the Control treatments; empty squares are the RDI 1, and triangles are the RDI 2 treatments. Each point is the average of 60 data. The vertical bar represents the standard error. The vertical solid line indicates the beginning of RDI 2 treatment. The vertical dash line indicates the beginning of RDI 1 treatment. Asterisks show when significant differences were found. (Tukey Test, $p < 0.05$).

Figure 9 shows the relationship between relative fruit volume, SWP (a) and SI (b). There was a very high scattering in both indicators. The enveloping curve used the percentile of 75% highest data in several intervals. These curves suggest a slight decrease of fruit size with water stress. In SWP data, values below −1.5 MPa tended to a clear reduction in relative fruit volume, but around −2.5 MPa this was still only 10%. Regarding SI data, the decrease of relative fruit volume started around 10 MPa day, but such decrease is also slow and values around 30 MPa day still reached around 90% of fruit volume.

Table 3 shows the yield data during the three seasons of the experiment. There were no significant differences in yield between treatments in any of the seasons. The highest yield was harvested in 2016, and the minimum was obtained in 2015, when it was almost null. In the 2014 season, the yield was around 15% lower than in 2016, but higher than the orchard average (8 t ha$^{-1}$). Control trees tended to produce a higher yield than those in deficits treatments in 2014 and 2016. In the 2014 season,

the Control trees production was around 18% higher than those in the RDI, which were almost equal. Such trend decreased in 2016, when the maximum differences were 9% between Control and RDI 1 trees, and almost null with RDI 2. Fruit size is one of the most important yield features for table olives. There were no significant differences in the number of fruits per kg between treatments in any of the seasons. The maximum size was measured in the 2015 season and the minimum in 2016 for all treatments. In 2014 and 2016, high fruit load seasons, Control trees tended to produce greater sizes than those with deficit irrigation, which were similar to each other. Such differences were similar in both seasons, between 9% and 12% and close to the values measured in the field (Figure 8). There were not significant differences in fruit load between treatments in any of the season. In 2014 and 2016, high yield season, Control trended to greater values and RDI 1 to the lowest, but such differences were only around 5%. In the 2015 season, RDI 2 presented the highest value of fruit load, almost double than Control, but there were no significant differences because of the high variability within treatments.

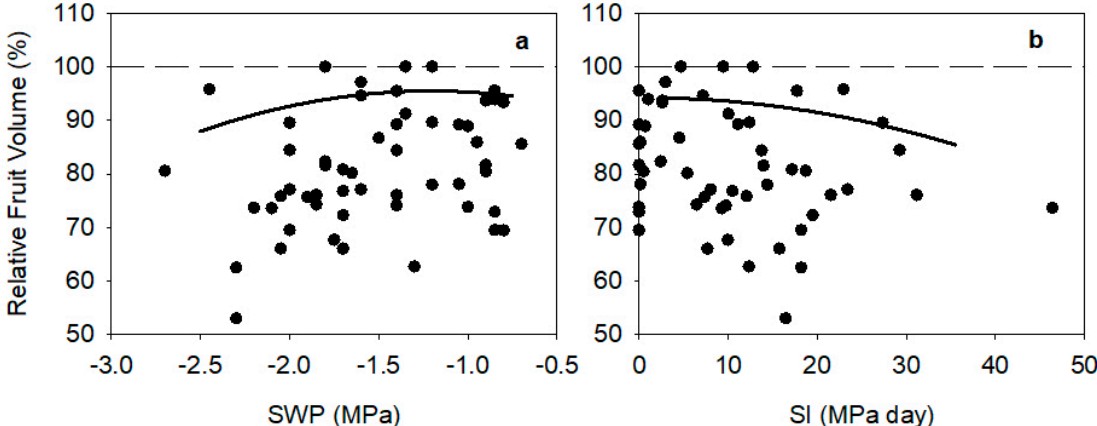

**Figure 9.** Relationship between relative fruit volume and midday stem water potential (SWP, **a**) and stress integral (SI, **b**). Each point is the data for an individual tree. The relative fruit volume was calculated considering 100% in each season as the greatest fruit volume from all the data available.

The pulp vs stone ratio (PS) is also a very important feature for table olives. There were significant differences in PS fresh weight in 2014 and 2015. In 2016 season, Control trees presented significantly larger PS ratio than the other two treatments. This trend was also measured in 2014 between Control and RDI 2. Conversely, the PS dry weight was significantly larger during 2015 in RDI 1 compared to Control, while in the other two seasons, the PS ratio in dry weight was almost equal for the different treatments. In the two seasons, when texture was measured (2015 and 2016), it was significantly larger in Control than in the two deficit treatments. Also related to this parameter, the mature index (MI) was significantly lower in 2015 in Control trees than in those with deficit treatments. In the 2014 and 2016 season, there were no clear trends, and the MI was below 1 in all treatments. Values of MI higher than 1 indicate a considerable number of fruits with black/purple spots related to the beginning of the ripening. Such spots reduce the yield value, and they were the reason for the early harvesting in 2015. The crop water productivity (CWP) was not significantly different between treatments in any of the seasons, but it was clearly larger in deficit treatments than in Control, mainly in RDI 2, where the minimum differences were 30% higher than in Control. This value was affected mainly by the amount of water used before the deficit period, because irrigation during treatments was higher in Control than in RDI 2 and intermediate in RDI 1, although not always significantly different.

**Table 3.** Yield quantity and quality features in the three years of the experiment. Yield (Y, t·ha$^{-1}$), Fruit size (S, fruits·kg$^{-1}$), Fruit load (FL, fruit tree$^{-1}$), Mature index (MI), Pulp stone ratio in fresh (PS F) and dry weight (PS D), Fruit texture (T, N), Water Applied per Season (AW, mm), Crop water productivity (CWP, kg·m$^{-3}$). Different letters in the same season and feature indicate significant differences ($p < 0.05$, Tukey Test).

| | 2014 | | |
|---|---|---|---|
| | **Control** | **RDI 1** | **RDI 2** |
| **Yield (T ha$^{-1}$)** | 14.7±1.6 | 12.2±2.4 | 12.0±2.4 |
| **Size (Fruit kg$^{-1}$)** | 244±9 | 261±21 | 275±15 |
| **FL (Fruit tree$^{-1}$)** | 12710±1656 | 11952±2953 | 11981±2922 |
| **MI** | 0.82±0.03 | 0.98±0.09 | 0.78±0.08 |
| **Pulp Stone Fresh** | 4.6±0.1 | 4.6±0.2 | 4.4±0.2 |
| **Pulp Stone Dry** | 2.1±0.1 | 2.2±0.1 | 2.2±0.1 |
| **Fruit texture (N)** | | | |
| **Applied Water (mm)** | 278±22a | 242±54ab | 143±13b |
| **CWP (kg m$^{-3}$)** | 5.1±0.3 | 5.0±1.0 | 8.3±1.5 |
| | **2015** | | |
| | **Control** | **RDI 1** | **RDI 2** |
| **Yield (T ha$^{-1}$)** | 0.2 ± 0.1 | 0.4 ± 0.1 | 0.5 ± 0.4 |
| **Size (Fruit kg$^{-1}$)** | 208 ± 4 | 192 ± 6 | 194 ± 4 |
| **FL (Fruit tree$^{-1}$)** | 144 ± 61 | 277 ± 110 | 348 ± 307 |
| **MI** | 0.87 ± 0.1b | 1.26 ± 0.1ab | 1.39 ± 0.1a |
| **Pulp Stone Fresh** | 5.8 ± 0.1 | 6.2 ± 0.1 | 5.7 ± 0.1 |
| **Pulp Stone Dry** | 2.5 ± 0.1b | 2.7 ± 0.1a | 2.5 ± 0.1ab |
| **Fruit texture (N)** | 5.9 ± 0.1a | 5.2 ± 0.2b | 5.2 ± 0.2b |
| **Applied Water (mm)** | 452 ± 28 | 409 ± 75 | 333 ± 33 |
| **CWP (kg m$^{-3}$)** | 0.1 ± 0.0 | 0.1 ± 0.1 | 0.1 ± 0.1 |
| | **2016** | | |
| | **Control** | **RDI 1** | **RDI 2** |
| **Yield (T ha$^{-1}$)** | 17.6 ± 1.8 | 16.1 ± 2.3 | 17.0 ± 2.0 |
| **Size (Fruit kg$^{-1}$)** | 309 ± 17 | 338 ± 10 | 331 ± 3 |
| **FL (Fruit tree$^{-1}$)** | 20,083 ± 4198 | 19,183 ± 4095 | 19,622 ± 556 |
| **MI** | 0.85 ± 0.1 | 0.71 ± 0.0 | 0.87 ± 0.1 |
| **Pulp Stone Fresh** | 4.8 ± 0.3a | 4.0 ± 0.2b | 3.9 ± 0.1b |
| **Pulp Stone Dry** | 1.8 ± 0.1 | 1.8 ± 0.1 | 1.8 ± 0.1 |
| **Fruit texture (N)** | 6.2 ± 0.2a | 5.0 ± 0.1b | 5.0 ± 0.1b |
| **Applied Water (mm)** | 457 ± 30 | 320 ± 68 | 295 ± 15 |
| **CWP (kg m$^{-3}$)** | 3.9 ± 0.8 | 5.5 ± 0.7 | 5.3 ± 0.3 |

## 4. Discussion

Accurate irrigation will need accurate information about water stress conditions. In these regards, threshold values or at least their estimation are very important to optimize limited water resources. Water status measurements are very useful for this purpose. The SWP baseline used in the present work [29] indicated that for most dates, the water status was close to the optimum (Figure 4). The current work validates this baseline in most of the data. But at the beginning of the season and during mid-summer in the high fruit load season, there were clear deviations (Figure 4). Reference [29] reported different equations according to fruit load but concluded that a unique baseline could be used. Decrease of SWP in olive trees like the ones reported in the current work during high fruit load season is reported in the literature [30,31]. Several fruit trees are including the use of water status with similar baselines in their irrigation management [32,33]. However, these baselines are not useful to manage water stress conditions, and threshold values are needed. During the deficit period in the current work, water stress level was not severe because leaf gas exchange was not clearly affected (Figures 4 and 6). Minimum midday stem water potential values around −2 MPa in the literature,

similar to the current work, are commonly considered moderate for olive trees, with no effect on yield [14] or even suggesting as no water stress conditions [15]. Such level of water stress could affect the current or the following season's yield. No effects were noticed on the yield of the following season (Table 3), because the vegetative growth occurred before the period of irrigation restriction and the number of fruits per shoot was similar in all treatments during the three seasons considered (Figure 7). Reference [34] reported that the alternate bearing in olive trees is mainly linked to shoot growth in the previous year. Reference [35] suggested in olive trees that floral induction could happen at the end of the summer period, but [14], considering a similar water stress level to the current work, reported no effect on the inflorescence number in the following season. Reference [36] also reported in olive trees the absence of effects on the flowering of the following season with irrigation restrictions in July and August, though unfortunately, no water stress level was reported in this latter work.

Current seasonal yield effects of water stress would affect the amount, size and colour of fruits, which are very important in table olive. There were no significant differences in colour in all seasons, except during the low fruit load one (Table 3). During this season, the main problem was early ripeness, because fruit spotting increased (Table 3). On the other hand, this response was not found in high fruit load seasons (2014 and 2016), although the levels and durations of water stress were clearly higher than in 2015 (Figures 4 and 5). Early fruit ripening has been related with water stress in some works in different olive cultivars (Arbequina, Ref. [37]; Carolea, Ref. [38]). However, this effect was not reported in cv Manzanilla in high fruit load seasons [14]. This reduction of the period of ripening is also described as a drought-avoiding response in plants [39]. Based on the current results, a low fruit load amplified this response in olive trees, even under mild water stress conditions, which did not occur in high fruit load seasons at least in cv Manzanilla. This earlier ripening is a very important factor to be considered in the irrigation for table olive production.

The fruit size is one of the main quality factors for table olives. No significant reductions in fruit size were found in the present work at harvest; however, there were significant differences in the fruit growth pattern and clear reduction trends around 10% (Figures 8 and 9, Table 3). Similar results in fruit size have been reported in other olive cultivars [40,41] but also greater decrease [38,42]. This effect of water deficit would associate with duration or level of water stress. The current work applied a fast and short water stress with minimum water potential around −2.5 MPa and a maximum SI around 30 MPa day (Figures 4 and 5). Figure 9 suggests that size reduction could start at around −1.5 MPa and 10 MPa day, slightly lower values have been suggested in the same and other cultivars, between −1.8 to −2.4 MPa [43–45]. However, this decrease was, sometimes, reversible at harvest. More severe water stress conditions than the current work (predawn or midday, lower than −4 MPa) clearly reduced the fruit size at the deficit period [40–42,45]. At yield, fruit size was near an optimum level or only 10% below in some works [40,41,45] but with a clear reduction after rehydration in others [42,45]. Fruit size recovery in [40,45] was reported even though there were partial rehydrations and SWP before harvest was around −2 MPa all time but not with values lower [45]. This threshold value is similar to the water stress level of the current work. However, this water stress level was not enough for a full recovery of fruit size in other works [41,42]. Such lack in the recovery was associated with a water stress during endocarp growth, which limited fruit growth even though water status was, in some treatments, optimum during recovery [42,46]. Water stress in the current work occurred after maximum endocarp size [17] and confirmed that SWP level around −2 MPa in this period would affect fruit size minimally.

## 5. Conclusions

Effects of water stress before harvest on yield quantity and quality were very limited. Yield quantity was not significantly affected for the irrigation treatments. Nevertheless, some quality parameters presented changes with water stress conditions. In low fruit load season, water stress level was mild, but it advanced ripening of the deficit treatments. This effect was not observed in high fruit load seasons. Fruit growth was sensitive to water deficit, but fruit size reduction was slow at moderate

water stress conditions. Midday stem water potential around −2 MPa was a successful threshold to reduce applied water with a minimum, most of the time no significant, effect on fruit size.

**Author Contributions:** Conceptualization, M.C., I.G. and A.M.; data curation, M.J.M.-P., M.C., I.G., L.A. and A.M.; formal analysis, M.J.M.-P., M.C., L.A., A.G., A.C., D.P.-L. and A.M.; Writing—Original draft, M.J.M.-P. and M.C.; Writing—Review and editing, A.M. All authors have read and agreed to the published version of the manuscript.

**Acknowledgments:** This research was supported by the Agencia Española de Investigación (AEI) and the Fondo Europeo de Desarrollo (FEDER) projects AGL2013-45922-C2-1-R and AGL2016-75794-C4-4-R.

**Conflicts of Interest:** The authors declare no conflict of interest.

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
