# Peer review of "Absence of Yield Reduction after Controlled Water Stress during Prehaverst Period in Table OliveTrees"

_agronomy, doi:10.3390/agronomy10020258_

Round 1

Reviewer 1 Report

The manuscript "Absence of yield reduction after controlled water stress during preharvest period in table olive trees" is an interesting study, well organized and conducted, with an extensive literature review, clear objectives and easy to read. It is an exhaustive study along three growing seasons that provides very useful information for the deficit irrigation scheduling in table olives, analysing not only the effect on the total yield, but also on other key parameters for table olives, such as the size of the fruit, the maturity index, etc.

The manuscript should be accepted in present form, but it is recommended that the authors paid special attention to the correction of small typographical errors such as double spaces or spaces between the number and the unit during final proof correction of the manuscript.

Author Response

We thank to this reviewer his/her kindly comments. We have revised the manuscript and changed some typographical errors.

Reviewer 2 Report

l 21: specify how many years you considered

l 27: in which period?

l. 29 eliminate .

l. 30 add olive

l. 51 consider also the effects on phenols see for examplehttps://doi.org/10.1371/journal.pone.0176580

l. 58-59 I suggest to eliminate this sentence since salt stress in the last phase has different mechanism of action compared to droughtstress

l.81 why this difference? Can this have affected the results? I think that this can create issues in comparing the 3 years ofexperimentation

l. 106-111 I suggest to resume in a table in order to be more clear

l. 112 is it a subsection?If yes give it a number and modify the subsequents subsection numbers

l. 149-154 why did you use a difefrent method? Specify

l. 167 specify the time point for the measurements

l. 167-168 specify how did you determine the fruit  volume

l. 236 whic type of problems? Can they have affected yourresults?

l. 376-377 eliminate bold

l. 399 discussion section: I suggest to avoid to repeat results and tomake a larger comparison of your data with the available literature

Round 2

Reviewer 2 Report

Line  58-59 please include:

https://doi.org/10.1371/journal.pone.0176580